# MMDuet2: Enhancing Proactive Interaction of Video MLLMs with Multi-Turn Reinforcement Learning

**Yueqian Wang**
Wangxuan Institute of Computer Technology, Peking University
`wangyueqian@pku.edu.cn`

**Songxiang Liu & Disong Wang & Nuo Xu & Guanglu Wan**
Meituan
`{liusongxiang, xunuo19, wangdisong, wanguanglu}@meituan.com`

**Huishuai Zhang** * **& Dongyan Zhao** *
Wangxuan Institute of Computer Technology, Peking University
National Engineering Research Center of New Electronic Publishing Technologies
`{zhanghuishuai, zhaodongyan}@pku.edu.cn`

## Abstract

Recent advances in video multimodal large language models (Video MLLMs) have significantly enhanced video understanding and multi-modal interaction capabilities. While most existing systems operate in a turn-based manner where the model can only reply after user turns, proactively deciding when to reply during video playback presents a promising yet challenging direction for real-time applications. In this work, we propose a novel text-to-text approach to proactive interaction, where the model autonomously determines whether to respond or remain silent at each turn based on dialogue history and visual context up to the current frame of a streaming video. To overcome difficulties in previous methods such as manually tuning response decision thresholds and annotating precise reply times, we introduce a multi-turn RL-based training method that encourages timely and accurate responses without requiring precise response time annotations. We train our model MMDuet2 on a dataset of 52k videos with two types of dialogues via SFT and RL. Experimental results demonstrate that MMDuet2 outperforms existing proactive Video MLLM baselines in response timing and quality, achieving state-of-the-art performance on the ProactiveVideoQA benchmark.

## 1 Introduction

In recent years, video multimodal large language models (Video MLLMs) have advanced rapidly. With increasingly sophisticated video understanding abilities and support for diverse input modalities (Li et al., 2024; Zhang et al., 2024b; Bai et al., 2025; Chen et al., 2024b; Zhang et al., 2024c; Xu et al., 2025), these models are being deployed across an expanding range of real-world applications.

Besides turn-based interaction where the model can only reply after the user's turn, proactive interaction has emerged as a promising and actively studied paradigm recently (Chen et al., 2024a; Wang et al., 2024; Qian et al., 2025; Yao et al., 2025). Proactive interaction is a more advanced requirement than online video conversation: it requires the model to not only understand interleaved visual and dialogue content, but also to determine on its own when to answer with appropriate content during the video playback. Achieving this requires continuous monitoring of visual and textual streams, real-time detection of salient events, and the ability to deliver timely, contextually relevant responses. Such proactive video MLLMs hold strong potential for real-time applications, including

---
*Corresponding Authors

live-stream analysis, intelligent surveillance, egocentric assistance agents, and socially interactive AI agents.

In previous works of proactive interaction (Chen et al., 2024a; Wang et al., 2024; Qian et al., 2025), a video MLLM determines whether it should respond after a certain frame by predicting response probability scores, such as using additional modules, the probability of a special token, or the visual token drop ratio, and compares the scores with a pre-defined threshold. However, there are two issues that remain:

(1) A threshold must be manually set during inference, and the model may never reply or often reply with duplicated content if this threshold is not set properly. To alleviate this problem, we use an entirely text-based approach to solve the problem of reply timing prediction: in each user turn, the user provides an optional textual content along with a small amount of visual information (1 or 2 frames from the online video), after which the assistant automatically initiates its own turn. The assistant can choose to output either a textual response or "NO REPLY" to indicate it does not want to reply right after this frame.

(2) Existing methods use supervised fine-tuning to train proactive interaction models, where exact reply timestamps for each model reply are required to construct training data, which is difficult to acquire as discussed later in Section 4.3.1. Prior studies typically insert responses either at the end of a scene or at a random position in the latter half of the scene to construct video-text interleaved dialogue data, followed by supervised fine-tuning during post-training. Due to difficulties in computational resources and data processing pipelines, scene segmentation is usually not too fine-grained, making it difficult to insert responses after the exact frame where the relevant visual information appears, which hinders the timeliness of model responses.

In this work, we leverage reinforcement learning to address this issue: with a niche reward design, we encourage the model to generate correct responses as early as possible while penalizing it for producing incorrect or excessively delayed responses. In this way we can significantly enhance the model's response timing without the need to annotate the precise timestamp of each model reply in the training data.

Using a carefully crafted proactive dialogue construction pipeline, we construct around 52k videos from YouTube and Ego-Centric videos, and two types of dialogue data: one question, multiple answers (1QnA) and multiple questions, multiple answers (nQnA). Based on this dialogue data, we trained MMDuet2 using SFT+RL, resulting in a state-of-the-art proactive video MLLM that achieves significant improvements over existing proactive model baselines in both response timing and response quality, and has state-of-the-art performance on ProactiveVideoQA.

In summary, the contributions of this work include: (1) An RL-based training method that can significantly improve proactive interaction experience without requiring precise reply timestamp in training data, (2) A pipeline for constructing proactive dialogue from videos, and a dataset consisting of 52k high-quality and diverse proactive dialogues, and (3) MMDuet2, a video MLLM that has state-of-the-art performance on proactive video QA benchmarks and provides a better interaction experience.

## 2 RELATED WORKS

### 2.1 PROACTIVE INTERACTION WITH VIDEOLLM

VideoLLM-Online (Chen et al., 2024a) is among the earliest efforts to adapt video–text MLLMs for proactive question answering. MMDuet (Wang et al., 2024) is trained on a wider range of tasks and datasets, achieving much better experimental results, but still struggles with issues such as inaccurate response timing and redundant outputs. Dispider (Qian et al., 2025) proposes a disentangled framework for proactive interaction separating perception, decision, and reaction, and TimeChat-Online (Yao et al., 2025) emphasizes token compression when processing input video streams. ProactiveVideoQA (Wang et al., 2025) proposes a comprehensive benchmark for proactive question answering evaluation, and proposes PAUC, an evaluation metric that is specially optimized for the fact that the model may give different responses at different times during proactive interaction.

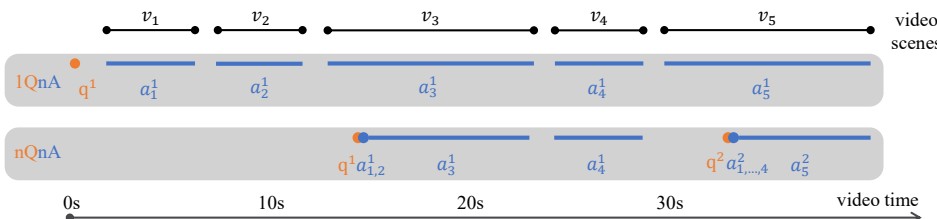

Figure 1: A conceptual demonstration of the proactive dialogues in the proposed dataset.

Many works focus on proactive reply in other tasks besides video question answering. LiveCC (Chen et al., 2025a) generates real-time video commentaries, Ego-Speak (Kim et al., 2025) studies speech initialization in face-to-face conversations. ViSpeak (Fu et al., 2025) focuses on recognizing body movements in the video to trigger specified responses, and (Panchal et al., 2024) studies providing timely feedback to fitness exercisers.

## 2.2 REINFORCEMENT LEARNING ON VIDEOLLM

Reinforcement learning has begun to play a transformative role in post-training video-text multi-modal language models, moving beyond purely supervised strategies. One of the prominent methods, VLM-RLAIF (Ahn et al., 2024) uses Reinforcement Learning from AI Feedback to align video and text representations by automatically generating self-preference feedback, bolstered by context-aware reward modeling that improves video grounding during instruction tuning. Another pioneer approach, Video-R1 (Feng et al., 2025), introduces a temporal extension to rule-based reinforcement learning T-GRPO, which explicitly incentivizes models to leverage correct frame order, helping better capture the temporal dynamics of video data. VideoChat-R1 (Li et al., 2025) further advanced this area by using Reinforcement Fine-Tuning with GRPO to boost spatio-temporal perception. LongVILA-R1-7B (Chen et al., 2025b) employs a two-stage pipeline, chain-of-thought SFT followed by RL, and uses sequence parallelism to extend RL training on long videos. R1-Omni (Zhao et al., 2025) integrates vision, audio, and language, adopts Reinforcement Learning with Verifiable Rewards (RLVR) and GRPO to train models that can recognize emotion in multimodal inputs. However, existing RL-enhanced VideoMLLMs have not explored real-time interaction or multi-turn dialogue, limiting their applicability in more interactive scenarios.

## 3 DATASET CONSTRUCTION

The videos of our proposed dataset contain two major categories: web videos and ego-centric videos. A dataset statistics is shown in Table 1. We use the following process to construct proactive QAs and their corresponding timespans in the video:

(1) **Scene segmentation and captioning.** Each video $V$ is first divided into a list of $n$ scenes $[v_1, v_2, \cdots, v_n]$, and we get a detailed scene caption for each scene: $[c_1, c_2, \cdots, c_n]$. We use different methods to obtain high-quality scene boundaries and captions for different categories of videos. For web videos from Live-WhisperX, as this dataset has good correspondence between video content and subtitles, thanks to its elaborate data cleaning process, we use the temporal boundaries of sentences in the subtitle as the boundary of the video scenes, and use frames sampled from this scene along with its subtitle as input to an MLLM to acquire a detailed caption. For ego-centric videos, as these datasets have detailed segment-level annotations, we directly use these annotations. We aim to segment videos into scenes that have relatively independent and clear video content, and these scenes occupy the vast majority of the time in the video, though they may not be connected end to end in time.

(2) **QA generation.** We use all scene captions as input, and instruct an LLM to generate a question $q$ and a list of $n$ answers $[a_1, \cdots, a_n]$, each $a_i$ corresponds to an scene $v_i$, and is derived from a scene caption $c_i$ that can answer the question $q$. If the information in $c_i$ can not answer $q$, then $a_i$ is set to "NO REPLY". We generate 2 to 4 question-answer lists for each video according to its video length and use superscripts $q^1, q^2, a_i^1, a_i^2$ to distinguish them in the rest of this paper.

| Type | #Videos | Video length | #Ques -tions | #Answer turns | Video source |
|------|---------|--------------|--------------|---------------|--------------|
| Web Videos | 50228 | 92.7 | 2.0 | 6.7 | Live-WhisperX (Chen et al., 2025a) |
| Ego Centic | 2543 | 164.4 | 2.1 | 5.6 | Ego-Exo4D (Grauman et al., 2023), EgoExoLearn (Huang et al., 2024) |

Table 1: Dataset Statistics.

(3) **Proactive dialogue construction.** We prepare 2 different types of proactive dialogues: "one question, multiple answers" (1QnA) and "multiple questions, multiple answers" (nQnA), each type covers half the number of all videos. A conceptual demonstration of the proactive dialogues is shown in Figure 1. In 1QnA dialogues, one question $q^j$ and one answer list $[a_1^j, \cdots, a_n^j]$ is used to construct one training example. The user asks the question $q^j$ at the beginning of the video, and the model should reply with an answer $a_i^j$ within the timespan of its corresponding video scene $v_i$. In nQnA dialogues, all of the 2-4 questions and answer lists are used to construct one dialogue. The user can ask any question $q^j$ at anytime. We use an LLM to summarize all answers of scenes that ends before the question time, i.e., $[a_1^j, \cdots, a_{t-1}^j]$ into one "immediate answer" $a_{\{1,\cdots,t-1\}}^j$, and the model should reply with this immediate answer at the time when the question is raised by the user. The answers of the following spans $[a_t^j, \cdots, a_n^j]$, they are still required to be replied within the corresponding scene until the next question is raised, then the model should then start to reply with the answers for the next question.

## 4 TRAINING PROCESS

### 4.1 FORMULATING PROACTIVE DIALOGUE WITH CHAT TEMPLATE

```
<|im_start|>system\nYou are a helpful assistant. Your task is
    to answer questions based on continuously incoming video
    frames. Your responses should include information from the
    video since your last reply (if any). If the information in
     this segment of the video cannot answer the question,
    output "NO REPLY".<|im_end|>
<|im_start|>user\n<image><image>What are people doing in
    office?<|im_end|>
<|im_start|>assistant\nNO REPLY<|im_end|>
<|im_start|>user\n<image><image><|im_end|>
<|im_start|>assistant\nPeople are working at desks with
    computers and monitors, engaged in various tasks.<|im_end|>
<|im_start|>user\n<image><image><|im_end|>
<|im_start|>assistant\nNO REPLY<|im_end|>
<|im_start|>user\n<image><image><|im_end|>
<|im_start|>assistant\nA reporter is speaking, people are busy
    at their desks with computers and monitors.<|im_end|>
```

Figure 2: Chat template of MMDuet2. User turns are marked in orange, assistant turns are marked in blue, and the textual contents of the dialogue between the two roles are underlined for the convenience of reading.

The chat template of proactive interaction we use is shown in Fig. 2. It proceeds in the following process:

1. First, we use a customized system message to indicate a proactive dialogue. This not only provides the model with the rules for its future responding, but also distinguishes proactive and offline video tasks with different contexts to reduce the catastrophic forgetting of offline understanding tasks during proactive training.

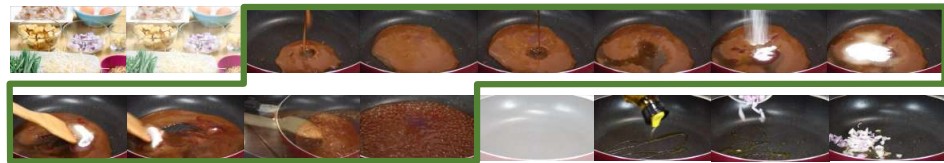

Figure 3: An example of a typical video snippet in dataset processing. Video frames circled by the green polygon constitutes a video scene.

2. The user inputs a message, which includes a few (1 or 2 in this paper) frames from the video, or a text input, or both frame and text.

3. In the assistant's turn, the model can choose to generate some text content as a reply, or generate "NO REPLY" to indicate that it does not want to reply in this round.

4. When the assistant's turn ends, the user retakes the floor and inputs a message containing frames or text. This loop continues until all sampled frames from the video have been input.

Within this chat template, the timestamp for each user turn or assistant turn in the video can be obtained by multiplying the number of frames preceding this turn by the time interval between consecutive frames. For instance, with a frame sample rate of 1 frame per second, the conversation in Figure 2 denotes the user says "What are the people doing in office?" at the 2nd second, the model replies "People are working..." at the 4th second and "A reporter is speaking..." at the 8th second.

Different from previous works like (Chen et al., 2024a; Wang et al., 2024), a major advantage of the chat template used in MMDuet2 is that it formats the entire interaction process, including video input, user input, reply time decision, and reply content generation, into messages from the user or the assistant and is therefore compatible with almost all popular post-training and inference frameworks. We know that there are more efficient strategies for reply time decision, but these methods require extensive modifications to the model architecture and code frameworks, which would take substantial labor to implement. We leave these discussions of potential methods for reply timing decision in the appendix.

## 4.2 SUPERVISED FINE-TUNING

We use Qwen2.5-VL 3B (Bai et al., 2025) as initialization. We hold out a few (1500 web and 400 ego-centric) videos for RL training, and use the rest of the dataset in the SFT training phase. The input frames are sampled at an interval of 2 seconds from the video and we use 128 tokens per frame, 2 frames per user turn. To build user-assistant conversations used in the SFT stage, we place model answers at the end of their reply timespans. This is to ensure that the relevant event has already occurred when making a reply (which is supposed to happen within the reply timespan, i.e., the corresponding video span) to avoid introducing hallucinations. To maintain the general offline video understanding abilities, we also include 25k offline video QA data from LLaVA-Video (Zhang et al., 2024d) and 25k video captioning data from tarsier2 (Yuan et al., 2025) in the SFT stage. These training examples are formatted using Qwen2.5-VL's default chat template and system prompt. This training process is conducted on 16 H800 GPUs and takes about 8 hours.

## 4.3 RL TRAINING

### 4.3.1 MOTIVATION OF USING RL

Although the model trained only with SFT has gained the ability for proactive responses, its performance is still unsatisfactory. We identified two main issues: First, the frequency of responses is relatively low. This may be because in the supervised training data, most of the turns are NO REPLY, causing the model to learn a bias towards this distribution. Second, the model often generates a response several seconds later than the key information appears, giving the user an experience of a long system delay. Automatically annotating ground-truth response time has been an unsolved challenge. For example, as shown in Figure 3, the caption for the scene circled by the green polygon is "*Tamarind, fish sauce and sugar are added to a heated pan and mixed using a spatula.*", which only

provides a coarse-grained scene-level annotation. It still remains unclear at which specific frame each ingredient appears so the model can generate a reply about this specific item in time. Pursuing overly fine-grained event timestamp would otherwise bring challenges in scene segmentation techniques and dataset construction cost.

Although providing accurate ground truth reply times is difficult, it is much easier to determine which of the two given proactive interaction outputs is better. An ideal proactive interaction system should generate replies both correctly (quantified by the text similarity between the model-generated answer and the ground truth answer) and early. More specifically, within each ground truth span, the preferences of the two responses should meet the following requirements: (1) With the same response time, the reply with higher correctness is more favorable; and (2) With the same increase in correctness induced by a new reply, the reply that comes earlier is more favorable. Therefore, an intuitive solution is to encourage the model to generate more favorable interactions through GRPO training with targeting rewards, circumventing the need to set ground truth response times. In this way, we can train the model to find the appropriate earliest response time by itself, since the model cannot generate a response about an event before actually observing it.

### 4.3.2 REWARD MODELING

Formally, let a video contain $G$ turns of ground-truth replies, where each turn consists of a textual response $gold_g$ and a corresponding reply timespan $(t_g^{start}, t_g^{end})$ for $g = 1, 2, \ldots, G$. This means that during the interval $(t_g^{start}, t_g^{end})$, the user expects to receive the information conveyed by $gold_g$. Within each reply timespan $(t^{start}, t^{end})$ (here we omit the subscript "$g$" for simplicity), a model $M$ generates $P$ model responses, each associated with a text $pred_p$ and a timestamp $\tau_p$, where $p = 1, 2, \ldots, P$ and $t^{start} < \tau_1 < \tau_2 < \cdots < \tau_P < t^{end}$. A correctness score $s_p \in [0, S]$ can be calculated upon each time the model generates a new reply (usually calculated by an LLM using the ground truth text and model response text as input), resulting in a list of correctness scores $s_1, s_2, \ldots, s_P$, where $S$ is a predefined max score.

The reward is inspired by the PAUC (Proactive Area Under Curve) (Wang et al., 2025) metric. A brief demonstration of the PAUC metric is shown in Figure 4. We plot the change in the model's response score over time as a polyline with $\tau$ on the x-axis and $s$ on the y-axis. In particular, we add a small score of 0.5 as the initial score at timestamp $t^{start}$, the reason behind this is that if a subsequent output gets a minimum score of $s = 0$, it will result in a worse PAUC metric than outputting nothing at all. PAUC is computed as the ratio between the area under this polyline and the maximum possible area:

$$PAUC = \frac{[(\tau_1 - t^{start}) \times 0.5 + \sum_{p=1}^{P-1} (\tau_{p+1} - \tau_p) \times s_p + (t^{end} - \tau_P) \times s_P]}{(t^{end} - t^{start}) \times S} \tag{1}$$

PAUC satisfies both of the above-mentioned requirements. As illustrated in Figure 4, increasing the score of a reply $((\tau_2, s_2) \rightarrow (\tau_2, s_2'))$ raises the height of the polyline on the y-axis, thereby yielding a larger area under the curve. If a reply can achieve a higher score than the previous one, the earlier this reply is made $((\tau_2, s_2) \rightarrow (\tau_2', s_2))$, the earlier the polyline rises on the y-axis, which also results in a larger area under the curve.

To better reflect the differences between responses in the reward and amplify the advantage of different rollouts for easier training, we made two minor modifications from the original implementation: we use max score $S = 4$ instead of 2, and each time when calculate the similarity between model reply and the ground truth, we only use the current turn of reply instead of all previous turns in the ground truth reply span in the original implentation of the PAUC metric. The modified PAUC reward is denoted as $r_{PAUC}$. Due to space limitations, for more details about PAUC please refer to its original paper.

Besides $r_{PAUC}$, we also use some additional reward to punish unwanted behaviors, mainly related to generating redundant and duplicate replies: (1) Replication reward ($r_{rep}$): To prevent the model from generating replicated replies as reported as a series problem in (Wang et al., 2024; 2025), and encourage the model to focus on new information in incoming videos, for each model reply, we use an LLM to judge whether all information in this reply is already covered in previous replies. We use the inverse of the ratio between the number of already-covered reply entries and the total number of

| | [WEB] | [EGO] | [TV] | [VAD] |
|---|---|---|---|---|
| VideoLLM-Online[†] | 25.9 / - | 25.0 / - | 18.3 / 53.9 | 25.0 / - |
| MMDuet | 38.9 / 81.3 | **46.0** / 99.4 | 21.1 / 92.8 | 27.4 / 99.2 |
| MMDuet2 $_{sft}$ (Ours) | 37.6 / **1.7** | 26.4 / **4.4** | 27.6 / 2.2 | 26.3 / **0.0** |
| MMDuet2 $_{rl}$ (Ours) | **53.3** / 4.2 | 33.6 / 8.1 | **43.4** / **1.0** | **28.9** / 15.2 |

Table 2: Performance on ProactiveVideoQA. Metrics reported are PAUC ($\omega = 0.5$) ↑/ reply duplicate proportion ↓, as defined in (Wang et al., 2025). [†]: Videollm-online generated more than 1 reply for only less than 10 answer turns on the [WEB], [EGO], and [VAD] datasets. Since the sample size is too small, we are not reporting this result as they have overly-large variance.

reply entries as $r_{rep}$. (2) In-span reward ($r_{in\_span}$): To prevent the model from generating replies during video spans unrelated to the question, we use the inverse of the ratio between the number of replies that do not fall in any ground truth reply span and the total number of reply entries as $r_{in\_span}$. (3) Prefix reward ($r_{pfx}$): We found that when generating new replies, sometimes the model may repeat the previous replies before adding new content, making the replies more verbose than necessary. To prevent this issue, for each turn of reply we calculate the longest common prefix between this reply and all previous replies, and mark the replies with the longest common prefix larger than a threshold as "verbose prefix reply". We use the inverse of the ratio between the number of verbose prefix replies and the total number of reply entries as $r_{pfx}$.

As shown in Eq. 2, we use $\omega_{PAUC}$, $\omega_{rep}$, $\omega_{in\_span}$, and $\omega_{pfx}$ to control the weights and use the weighted sum of the 4 rewards as the overall reward:

$$r = \omega_{PAUC} \times r_{PAUC} + \omega_{rep} \times r_{rep} + \omega_{in\_span} \times r_{in\_span} + \omega_{pfx} \times r_{pfx} \quad (2)$$

The first term $\omega_{PAUC} \times r_{PAUC}$ is more likely to assign higher reward to samples with more reply turns, while the latter terms $\omega_{rep} \times r_{rep} + \omega_{in\_span} \times r_{in\_span} + \omega_{pfx} \times r_{pfx}$ are more likely to assign higher reward to samples with less rewards, forming a tradeoff between informativeness and simplicity. In practice, we found that a good weighting scheme should make the influence of the former slightly stronger than that of the latter. This allows the model to gradually shift from the low-frequency, high-latency replies after solely training with SFT, toward generating more frequent and timely proactive replies, while without resorting to reward hacking $r_{PAUC}$ by producing a large number of redundant replies. After some hyperparameter search we find that $\omega_{PAUC} = 3, \omega_{rep} = 2, \omega_{in\_span} = 0.5, \omega_{pfx} = 2$ is good and use these hyper-parameters in the subsequent experiments.

### 4.3.3 TRAINING DETAILS

Since a complete video corresponds to multiple ground truth reply spans, performing rollout over the entire video and providing only a single averaged reward would result in very sparse rewards. Moreover, as the rewards for different ground truth reply spans are computed independently, using an average reward will introduce a temporal credit assignment problem (Sutton, 1984), making it difficult to attribute the final reward to specific ground truth reply spans. To alleviate this problem, in each step we only select a short span (from 20 to 60 seconds) from the video for training and provide ground truth model replies for the dialogue turns that happen before the selected span. We sample frames with an interval of 2 seconds from the video and use 128 tokens per frame, 2 frames per user turn. We use GRPO (Shao et al., 2024) with a number of rollouts as 4, implemented with SGLang (Zheng et al., 2024) and verl (Sheng et al., 2025) framework. This training process is conducted on 8 H800 GPUs and takes about 20 hours.

## 5 EXPERIMENTS

### 5.1 EXPERIMENTS ON PROACTIVE BENCHMARKS

Performance on existing benchmarks, ProactiveVideoQA (Wang et al., 2025) and StreamingBench proactive output (Lin et al., 2024) task (**PO**), are listed in Table 2 and 5. Given the deployment of proactive interaction can be very complex, reproducing without the official proactive inference code

|  | # Reply Turns | Wall Time |
|---|---|---|
| MMDuet | 5.7 (3.4) | 2m27s |
| MMDuet2 | 3.3 (1.9) | 2m52s |

Table 3: Inference Wall Time on `[WEB]`.

|  | Video-MME (w/wo sub) | MVBe-nch | LongVid-eoBench |
|---|---|---|---|
| Qwen2.5-VL 3B | 67.6/61.5 | 67.0 | 54.2 |
| Qwen2.5-VL 3B[†] | 66.5/57.3 | 65.6 | 53.1 |
| MMDuet2 $_{sft}^{†}$ (Ours) | 67.1/57.7 | 65.3 | 53.3 |
| MMDuet2 $_{rl}^{†}$ (Ours) | 67.5/58.1 | 66.4 | 52.7 |

Table 4: Performance on several popular offline video understanding benchmarks. [†]: Our implementations.

|  | Acc |
|---|---|
| VideoLLM-Online | 1.96 |
| Dispider | 25.34 |
| MMDuet | 29.44 |
| MMDuet2 $_{sft}$ (Ours) | 19.59 |
| MMDuet2 $_{rl}$ (Ours) | **34.69** |

Table 5: Performance on Proactive Output task of Streaming-Bench.

|  | [WEB] | [EGO] |
|---|---|---|
| MMDuet2 | 53.3/4.2/3.3 | 33.6/8.1/3.5 |
| $-r_{rep}$ | 55.5/17.3/4.9 | 35.6/31.9/8.0 |
| $-r_{pfx}$ | 53.0/4.3/3.1 | 27.5/2.3/0.6 |
| $-r_{in\_span}$ | 62.7/9.6/8.4 | FAIL* |

Table 6: Ablation studies on each reward item. Metrics reported are PAUC↑/ reply duplicate proportion ↓/num reply turns. Adverse consequences caused by removing a reward item are marked in red. *FAIL: The model generates response at almost every turn, regardless of whether truly relevant to the question. Evaluation on this task is unfeasible as inference on a single data point can take more than 20 minutes.

could lead to suboptimal results, potentially leading to misunderstandings about the capabilities of those models. So we compare with MMDuet (Wang et al., 2024) and VideoLLM-Online (Chen et al., 2024a) as only these two models have open-sourced code for proactive evaluation. As the videos in `[WEB]` of ProactiveVideoQA and **PO** of SteamingBench are relatively short in length (16.59 and 13.14 secs respectively), we use 1 frame per user turn during inference. For the other three tasks (`[EGO]`, `[TV]`, and `[VAD]`) with longer videos, we use 2 frames per user turn.

Results show that MMDuet2 outperforms existing proactive interaction models by a large margin. Although MMDuet achieves good performance on tasks such as `[EGO]`, it achieves so by generating a large number of unnecessary and repetitive replies to fulfill the goal of "conveying useful information as early as possible". However, all models perform relatively poor on `[VAD]`, demonstrating that current models still struggle to understand surveillance videos. Some real proactive dialogue case videos can be found in the supplementary material.

**Inference Speed.** Here we report the actual inference speed (wall time) of the `[WEB]` task for MMDuet and MMDuet2. We take the following measures to ensure the comparison is as fair as possible: We use the same computational node with an H100 GPU while maintaining GPU utilization at 100%, select 64 samples from the ProactiveVideoQA `[WEB]` task and test the inference wall time. Results shown in Table 3 demonstrate that inference time is comparable though performing a complete generate operation to produce "NO REPLY" at every decision.

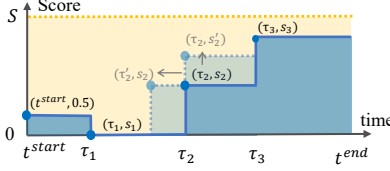

Figure 4: An illustration of the calculation of the PAUC metric (Wang et al., 2025).

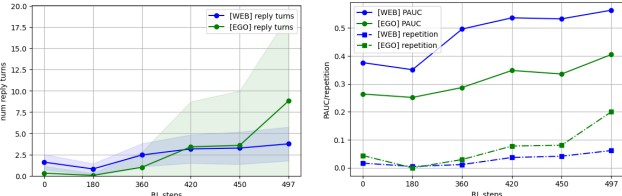

Figure 5: Dynamics of key metrics of model behavior during RL training.

| | | [WEB] | | | [EGO] | |
|---|---|---|---|---|---|---|
| SFT frame interval | 1 sec | 2 secs | | | 2 secs | |
| RL frame interval | - | 1 sec | 2 secs | 1 sec | 2 secs | |
| Inference | 1 sec | FAIL | 47.0/1.0 | 53.3/4.2 | 34.7/3.9 | 33.6/8.1 |
| frame interval | 2 secs | FAIL | 39.4/0.0 | 44.2/1.7 | 30.6/1.8 | 33.5/7.5 |

Table 7: Performance of using different frame interval for SFT, RL and inference.

## 5.2 Experiments on Offline Video-Text Benchmarks

To verify whether our training introduces any negative impact on offline video understanding tasks, we also report performance on several widely used offline video understanding benchmarks: Video-MME (Fu et al., 2024), MVBench (Li et al., 2023), and LongVideoBench (Wu et al., 2024). When testing on these offline video understanding benchmarks, we use the default system prompt of Qwen2.5-VL ("You are a helpful assistant.") instead of our customized system prompt stated in fig. 2. We also report the evaluation results of our implementations, which use lmms-eval framework (Zhang et al., 2024a) for evaluation and set max tokens per frame as 256. This is to ensure a fair evaluation of the impact of the post-training process proposed in this paper, as it has been reported that there are big gaps between the reproduced results using lmms-eval and the performance reported in the original paper (Bai et al., 2025). Results are shown in 4. After fine-tuning and reinforcement learning for enhancing proactive interaction, MMDuet2's performance on offline video understanding benchmarks remains almost the same as the checkpoint before our post-training.

## 5.3 Ablation Studies

**Impact of Each Reward Item.** We demonstrate the necessity of $r_{rep}$, $r_{in\_span}$ and $r_{pfx}$ in Table 6. Results show that $r_{rep}$ and $r_{in\_span}$ are indispensable: without any of these 2 rewards, the model generates more duplicated responses to achieve an unreasonably high PAUC metric. Moreover, without $r_{in\_span}$, the model significantly increases the response density, which makes it unfeasible to evaluate on long videos in [EGO].

The impact of $r_{pfx}$ is more complicated. During training, we observed an undesirable pattern where the model would first generate verbatim repetitions of the previous response before continuing to generate new content. Experimental results also show that without this reward, the model may fail on the more difficult [EGO] test set. Therefore, we add this reward as a penalty.

**Impact of frame sample density in training and inference is different.** The frame sampling density during training and inference can be a potentially critical factor influencing the interactive experience. We experimented with different frame intervals during SFT, RL, and inference phases, and the experimental results are shown in Table 7. In SFT phase, when frame interval is set to 1 second, the model will collapse to always generating "NO REPLY" in every reply turn, as the training data is overly biased towards not replying, so mark these results as "FAIL" in Table 7, and use 2 sec frame interval for SFT in subsequent experiments. In the RL phase, we found that setting different frame intervals does not have a significant impact on performance. However, in the inference phase, we found that reducing the frame interval from 2 seconds to 1 second leads to a significant performance improvement. The underlying reason is that a lower frame interval leads to a higher decision rate, allowing the model to perceive the appropriate response timing earlier, which is more favorable both for user experience and for the PAUC metric. This also demonstrates the robustness of MMDuet2 to different frame sampling strategies.

We observe that the proactive interaction training framework proposed in this work demonstrates strong generalization performance with respect to the frame sampling interval. Therefore, we recommend using a relatively low-frequency training frame rate (2 seconds per frame) to save training costs, and a relatively high-frequency inference frame rate (1 second per frame) to achieve a better interaction performance.

## 5.4 Training Dynamics of the RL Process

In this subsection, we aim to present the changes in model behavior during the RL training process. In Figure 5, we show line charts of several key metrics during proactive video QA testing: the model's average number of response turns, PAUC score, and repetition. From the line chart we can clearly identify that the RL process can be divided into three stages:

Stage 1 (step 0–180) shows a transition period, both the number of responses and the performance decline. As the training objectives and encouraged response patterns of SFT and RL differ, the model is in the process of switching from the old to the new response pattern during this stage, which leads to a slight drop in performance.

Stage 2 (step 180–450) shows a growth period, the model, guided by the training objective, learns to generate earlier and more accurate responses while avoiding overly frequent or repetitive replies. During this stage, the model's response frequency and PAUC performance increase rapidly, and repetition remains generally controllable.

Stage 3 (step 450–489) shows a plateau period, the model's performance on the [WEB] network video task stabilizes. However, on the [EGO] ego-centric video task which is longer and more challenging for content understanding, the model can have some generalization issues as we observe an increase in repetition. We believe this can be alleviated by collecting more ego-centric and long-form videos as training data, which will be an important direction for future work.

## 6 Conclusion

Moving beyond conventional user-turn-initiated conversation paradigms, in this paper, we studied improving proactive interaction of video multimodal large language models, which enables the model to autonomously decide when to respond during video playback. We constructed a large-scale proactive dialogue dataset comprising 52k videos with diverse question-answer structures, facilitating robust training. We proposed a method to represent the proactive interaction process solely using messages between the user and the assistant without modifying the model structure, which allows our model to be directly adapted to most training and inference frameworks, facilitating hands-on usage.

By integrating reinforcement learning with a specialized reward design, we train MMDuet2, a proactive Video MLLM that significantly enhances the correctness and timeliness of proactive dialogue while reducing redundant and repetitive replies. MMDuet2 demonstrates superior performance over existing baselines on several proactive output tasks without having degraded performance on offline video understanding benchmarks.

Future research directions include: (1) Collecting more diverse data, such as teaching and learning processes, to train a more versatile proactive Video MLLM that goes beyond solely QA, (2) Integrating with techniques like visual token compression to cut down on computation expenses, and (3) Combining with speech understanding and generation to extend proactive interaction capabilities to more modalities.

### Acknowledgments

This work is supported by National Science Foundation of China (No. 62576016, 62576015) and Beijing Natural Science Foundation (L253001).

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

## A DISCUSSION OF REPLY TIMING DECISION METHODS

Here we describe a more efficient implementation of reply timing instead of generating "NO RE-PLY" as described in Section 4.1. In most MLLMs including Qwen2.5-VL, visual tokens are surrounded by special tokens like: `<vis_start><vis_token>...<vis_token><vis_end>`. Given the content up to the last `<vis_end>`, we can train the model to predict whether the next token is `<vis_start>`, indicating that the model wants to see one more piece of visual information, or `<im_end>`, indicating that the model want to stop the user's turn and start its own turn. In this format, if the model chooses not to respond, no additional token will be added to the context, ensuring high information density. However, this requires introducing new rules into inference frameworks like SGLang (Zheng et al., 2024) or vLLM (Kwon et al., 2023), which requires significant labor. Therefore, we leave utilizing this more token-efficient format as future work.

## B    USE OF LARGE LANGUAGE MODELS (LLMS)

LLM-based tools are used to assist in writing the experimental code. LLMs are also employed to translate parts of the manuscript into English or to polish the text without altering its semantics.

