# OpenReview forum: "MMDuet2: Enhancing Proactive Interaction of Video MLLMs with Multi-Turn Reinforcement Learning"
_ICLR.cc/2026/Conference — ICLR 2026 Poster_

### Official Review · Reviewer_3Vfh · 2025-10-30

**Soundness:** 3
**Presentation:** 2
**Contribution:** 2
**Rating:** 4
**Confidence:** 4

**Summary:**

This paper introduces a novel video multimodal large language model (Video MLLM) called MMDuet2, designed to enhance the model's proactive interaction capabilities—that is, the ability to autonomously decide when and how to respond while watching streaming videos. To address the challenges of existing methods requiring manual threshold adjustment and precise timestamp annotation of responses, the authors propose an innovative multi-round reinforcement learning (RL) training method. This method encourages the model to make timely and accurate responses at the right time through a carefully designed reward mechanism, without the need for precise timestamp annotation. Furthermore, the authors constructed a large-scale proactive dialogue dataset containing 52k videos for training. Experimental results show that MMDuet2 achieves state-of-the-art performance on benchmarks such as ProactiveVideoQA, significantly outperforming existing models.

**Strengths:**

1. This paper innovatively introduces multi-round reinforcement learning, using a reward mechanism to teach the model to find the optimal response time, cleverly circumventing the challenge of precise time labeling.
2. The authors constructed a large-scale dataset containing 52k videos, providing a solid data foundation for training more robust active models.
3. Experimental results show that the MMDuet2 model trained with SFT+RL outperforms previous state-of-the-art models and our own model trained solely with SFT on authoritative active interaction benchmarks such as ProactiveVideoQA. This demonstrates the effectiveness of reinforcement learning methods.

**Weaknesses:**

1. Using the "NO REPLY" text token is a concise and universal approach, but it also means that if the model chooses not to respond, a complete generation process (generating both tokens) is still required, which limits its inference efficiency.
2. The study on the reward component is insufficient, and related ablation experiments are lacking. The total reward is a weighted sum of four components, but the paper only mentions "a certain hyperparameter search" providing a set of weights without conducting ablation studies. This fails to clearly explain the specific contribution of each reward or penalty to the model's performance.
3. The task is limited to question-and-answer type tasks, and from data construction to model evaluation, it relies heavily on the question-and-answer paradigm. Therefore, the model's generalization ability for non-question-and-answer type proactive response tasks, such as caption tasks, needs to be tested.
4. There is insufficient comparative testing with other active response models. Related work mentions some active response models, such as Dispider [2] and TimeChat-Online[3]. However, the main experiments did not directly compare these models.

[1] Joya Chen, Zhaoyang Lv, Shiwei Wu, Kevin Qinghong Lin, Chenan Song, Difei Gao, Jia-Wei Liu, Ziteng Gao, Dongxing Mao, and Mike Zheng Shou. Videollm-online: Online video large language model for streaming video. 2024 IEEE/CVF Conference on Computer Vision and Pattern Recognition (CVPR), pp. 18407–18418, 2024a.

[2] Rui Qian, Shuangrui Ding, Xiao wen Dong, Pan Zhang, Yuhang Zang, Yuhang Cao, Dahua Lin, and Jiaqi Wang. Dispider: Enabling video llms with active real-time interaction via disentangled perception, decision, and reaction. ArXiv, abs/2501.03218, 2025.

[3] Linli Yao, Yicheng Li, Yuancheng Wei, Lei Li, Shuhuai Ren, Yuanxin Liu, Kun Ouyang, Lean Wang, Shicheng Li, Sida Li, Lingpeng Kong, Qi Liu, Yuanxing Zhang, and Xu Sun. Timechatonline: 80% visual tokens are naturally redundant in streaming videos. ArXiv, abs/2504.17343, 2025.

**Questions:**

1. The dialogue template shown in Figure 2 of the paper appears to be standard practice for frame-by-frame streaming video understanding models. Furthermore, the core mechanism for generating "NO REPLY" seems functionally equivalent to predicting a specific EOS token (VideoLLM-online [1]). Therefore, aside from the specific implementation, what is the fundamental innovation of this "text-to-text approach"?
2. To better contextualize the performance of MMDuet2, would the authors consider providing direct empirical comparisons against other relevant models, such as Dispider [2] and TimeChat-Online [3]? At the same time, could you please provide an ablation study analyzing the individual impact of the four reward components ($r_{\text{PAUC}}$, $r_{\text{rep}}$, $r_{\text{inspan}}$, $r_{\text{pfx}}$) on the model's behavior?
3. The authors employed a strategy of placing the model's response at the end of its response time period during the SFT phase to avoid the model developing illusions before seeing the relevant event. However, the goal of the RL phase is to encourage the model to make the correct response as early as possible. Are these two goals contradictory during training?

---

> ### Author Response · Authors · 2025-11-19
> **Author Response to Reviewer 3Vfh**
>
> Thank you for your suggestions to our work!
>
> **W1: Generate "NO REPLY" limits inference efficiency.**
>
> We have discussed this in Lines 232-235, and in Appendix A. We proposed a better solution that does not require generating additional tokens. The choice of generating "NO REPLY" was solely constrained by the feasibility of the implementation.
>
> **W2 & Q2: Ablation study on the reward.**
>
> As per your request, we have added an ablation study where we remove $r\_{rep}$, $r\_{in\\_span}$, and $r\_{pfx}$ to verify the necessity of these three rewards. The results are as follows:
>
> |                     | [WEB]  |   | [EGO]   |  |
> | --------------| -------- | ----------- | ----------- |  ----------- |
> |                    | PAUC/duplicate| num turns | PAUC/duplicate | num turns |
> | MMDuet2              | 53.3/4.2  | 3.3       | 33.6/8.1 | 3.5 |
> | \$-r\_{rep}\$        | 55.5/17.3 | 4.9       | 35.6/31.9 | 8.0 |
> | \$-r\_{pfx}\$        | 53.0/4.3  | 3.1       | 27.5/2.3 | 0.6 |
> | \$-r\_{in\\\_span}\$ | 62.7/9.6 | 8.4       | FAIL\* |     |
>
> \*FAIL: The model generates response at almost every turn, regardless of whether truly relevant to the question. Evaluation on this task is unfeasible as inference on a single data point can take more than 20 minutes.
>
> **This new ablation study confirms that \$r\_{rep}\$ and \$r\_{in\\\_span}\$ are indispensable:** without any of these 2 rewards, the model generates more duplicated responses to achieve an unreasonably high PAUC metric. Moreover, without \$r\_{in\\\_span}\$, the model significantly increases the response density, which makes it unfeasible to evaluate on long videos in [EGO].
>
> **The impact of \$r\_{pfx}$ is more complicated.** During training, we observed an undesirable pattern where the model would first generate verbatim repetitions of the previous response before continuing to generate new content. Experimental results also show that without this reward, the model may fail on the more difficult [EGO] test set. Therefore, we add this reward as a penalty.
>
> **W3: Test on non-QA tasks.**
>
> Thank you for your suggestion. However, based on the current available open-source resources, it is only feasible to evaluate proactive interaction models on QA tasks. Also, many QA examples with not-too-specific questions (e.g., "What is happening in the office?") are almost equivalent to dense captioning, so our experimental results can reflect captioning abilities to some extent.
>
> **W4 & Q2: Comparing with more baselines like Dispider and Timechat-Online.**
>
> As stated in Line 370, **we do not compare with these baselines because ​they have not released their proactive inference code**​, as shown in this issue: [https://github.com/Mark12Ding/Dispider/issues/3](https://github.com/Mark12Ding/Dispider/issues/3). We also reached out to the authors of Timechat-Online, and they currently have no plans to release proactive interaction code. Given the deployment of proactive interaction can be very complex, reproducing without the official proactive inference code could lead to suboptimal results, potentially leading to misunderstandings about the capabilities of those models..
>
> **Therefore, we only compared with baselines that have publicly available proactive interaction code.** We will clarify this in our manuscript.
>
> **Q1: What is the fundamental innovation of using the "text-to-text approach" to represent proactive interaction?**
>
> Our fundamental improvements are:
>
> 1. The model architecture and chat template is identical to Qwen2.5-VL, making it compatible with almost all training and inference frameworks (Lines 232-235).
> 2. No threshold is required during inference, addressing a limitation that has not been resolved by any of the existing proactive interaction works (Lines 48-55).
>
> **Q2: Can you provide direct comparisons with relevant models and an ablation study on the reward components?**
>
> See W2 & W4.
>
> **Q3: Are the training objectives of the SFT and RL phases contradictory? ​**
>
> **Yes. Theoretically, the delayed responses during the SFT phase and the desire for earlier responses during the RL phase may conflict.** However, as we summarized in Lines 60-63, in cases where finer-grained data processing is difficult to implement, we believe it is more important to avoid introducing hallucinations, even if it means compromising on response timing. This conclusion is also derived from our analysis of the MMDuet paper.
>
> We hope the above responses address your concerns and provide a clearer understanding of our manuscript. We would greatly appreciate it if you could consider increasing your rating.

---

### Official Review · Reviewer_rgJo · 2025-11-01

**Soundness:** 3
**Presentation:** 3
**Contribution:** 3
**Rating:** 4
**Confidence:** 3

**Summary:**

This paper proposes MMDuet2, aiming to enhance the proactive interaction capabilities of Video Multimodal Large Language Models (Video MLLMs) through multi-turn Reinforcement Learning (RL), enabling the model to autonomously decide when to respond during video playback. The authors construct a new dataset of 52k videos and present a text-to-text RL approach with a reward mechanism, including PAUC.

**Strengths:**

1.The paper addresses proactive interaction, which is an important and challenging promblem for making Video MLLMs more natural and practical in real-time applications.

2.  The use of RL to overcome the difficulty of precise reply time annotation is a promising avenue, and the reward mechanism design  theoretically considers timeliness, accuracy, and redundancy.

3. The creation of a large-scale new dataset (52k videos with two dialogue types) provides a valuable resource for research in this field.

**Weaknesses:**

1. The central contribution of this paper lies in rl , which is explicitly designed to improve proactive interaction timing. However, the paper reports that during training on complex ego-centric video tasks, the model exhibited reward hacking behavior—generating large amounts of repetitive content. Although this issue is solved by early stopping, such manual action may show a instability in the reward design and optimization process.

2. The occurrence of reward hacking indicates that the learned policy may not genuinely capture proactive interaction behavior but instead exploits the reward function. The paper does not provide ablations or diagnostic analyses to clarify why this failure occurs, nor does it offer evidence that the method can generalize to more complex or long-duration scenarios without collapsing.

**Questions:**

1. Regarding the observed reward hacking issue, was there a deeper analysis of its root causes beyond "early stopping"? Were more sophisticated reward shaping, curiosity mechanisms, or RL algorithms attempted to enhance the model's robustness on complex tasks?

2. Please elaborate on how the largely "offline" QA and captioning datasets in the SFT stage were processed and transformed to effectively support RL training for proactive interaction, rather than merely enhancing general understanding?

3. The paper states that reducing the frame interval from 2 seconds to 1 second during inference significantly improves performance, even if the RL phase used a 2-second interval. Does this suggest that the model's decision-making process is highly sensitive to the temporal granularity of the input?

---

> ### Author Response · Authors · 2025-11-19
> **Author Response to Reviewer rgJo**
>
> Thank you for your suggestions to our work!
>
> **W1: Repetitive content on ego-centric videos.**
>
> ​**This is because videos in the [EGO] task is ego-centric and relatively long ​**​(360s, compared to the average of 164s in our training data)**. ​**Such videos are difficult to collect in large quantities as training data, **so the data distribution of this task significantly differs from our training data.** As a result, we observed that the model produces excessive responses for some test examples, and faithfully reported this issue in the paper.
>
> We believe that this limitation should not serve as grounds for rejection, as this issue can be alleviated if more  long-form ego-centric training data are collected, which is an important direction for future work. **This does not undermine the core contribution of the paper,** which is to demonstrate that the proposed data construction, SFT, and RL methodology improves the model's proactive interaction capability.
>
> **W2 & Q1: The issue of "reward hacking".**
>
> You may be referring to "the model attempts to hack the reward by generating large amounts of repetitive content" in Line 462. **This is a miswording: what we actually intended to express was "a generalization issue".**
>
> **Reward hacking** refers to situations where "the reward function is not well-aligned with the real objective, and during the RL training process, the agent may find ways to receive high rewards without actually solving the task". **However, this is actually not our case.** As shown in Table 2, we did not observe a significant increase in repetition after RL training on tasks with data distributions similar to the training data (e.g., [WEB], [TV]). We only observed a noticeable increase in repetition on the more challenging long-form ego-centric videos and surveillance videos (e.g., [EGO], [TV]). **Therefore, this is actually a generalization issue caused by the difficulty in collecting certain types of training data, and the model's understanding ability on these videos still needs improvement. ​**We apologize for the miswording and will correct it in the manuscript.
>
>
> **Q2: Does offline datasets support proactive interaction?**
>
> **Offline video QA data is not intended to enhance proactive interaction, and they are only used in SFT (not RL). ​**The motivation for introducing offline QA data is to maintain general understanding capability, which is a common practice in many LLM training works.
>
> **Q3: Is the model's decision-making process highly sensitive to the temporal granularity of the input?**
>
> **No. Reducing the inference frame interval from 2 seconds to 1 second benefits the model by enabling it to detect the appropriate time for a response 1 second earlier. ​**This is absolutely more favorable, both in terms of  evaluation metric and the actual user experience (as it reduces the delay).
>
> We hope the above responses address your concerns and provide a clearer understanding of our manuscript. We would greatly appreciate it if you could consider increasing your rating.

---

### Official Review · Reviewer_Rz9J · 2025-11-01

**Soundness:** 4
**Presentation:** 2
**Contribution:** 3
**Rating:** 6
**Confidence:** 3

**Summary:**

This paper investigates the reward-based RL in online video LLM settings and propose MMDuet2 to autonomously determine whether to respond as soon as possible or remain silent in a proactive manner by RL training. This paper starts from curating online video llm training data, which is designed for multiturn proactivate dialogue, then design the specialized chat template for it and use SFT and RL to train the model in the off-policy and on-policy way for proactivate capabilities. Experimental results show that the proposed MMDuet2 can proactively answer user's queries in the online video streaming setting while maintain the ability to answer in offline video setting.

**Strengths:**

1) This paper is one of the first batch to investigate RL training (esp. GRPO) for proactively answering in online streaming video settings, which is of substantial novelty. Also, the paper investigate the reward, the key component of GRPO, and model it specifically for online video settings (PAUC).

2) The authors design a training dataset especially for online video streaming setting and corresponding chat template. Looking forward to the dataset open sourcing.

3) The MMDuet2 trained by SFT and GRPO outperform previous online methods while maintaining offline video understanding capabilities

**Weaknesses:**

There are no major technical concerns about this paper, but I want to address some minor points as follows:

1) As the key component to apply GRPO to online video settings, the ablations on rewards should be more addressed. Did authors try other rewards than PAUC? Please compare several reward formulations and discuss why PAUC is preferred.

2) The author is encouraged to report the actual inference speed and latency of the MMDuet2 to see if it is realtime in practical scenes.

Also, the organization of this paper needs to be improved further since the current version seems too flattened with every details being straight written without a main storyline.

**Questions:**

See weaknesses.

---

> ### Author Response · Authors · 2025-11-19
> **Author Response to Reviewer Rz9J**
>
> Thank you for your suggestions to our work!
>
> **W1: Did we try rewards other than PAUC?**
>
> **Yes, we made several modifications to PAUC when selecting the reward and used the best-performing modified version as the final reward.** This process is outlined in Lines 305-308. The reason we can not list the entire modification of the reward based on PAUC in the ablation study is that the modifications were tested on a subset before the entire training dataset is prepared, and its infeasible to re-run the experiments for ablation study results.
> The reason for choosing PAUC as the base metric for modification is it is the only metric that is intuitively reasonable and has been validated through agreement with human preference.
>
> **W2: Actual Inference Speed.**
>
> **We have added an experiment on the actual inference speed (wall time) in the [WEB] task.** We use the same computational node with an H100 80G GPU while maintaining GPU utilization at 100%, select 64 samples from the ProactiveVideoQA [WEB] task and test the inference wall time. The results are as follows.
>
> | Model              | avg. (std.) rep turns | Wall Time |
> | -------------------- | ----------------------- | ----------- |
> | MMDuet             | 5.7 (3.4)             | 2min27s   |
> | MMDuet2\_rl (Ours) | 3.3 (1.9)             | 2min52s   |
>
> We initially did not report this result because the actual inference speed is influenced by many factors, such as hardware, operator optimization, model parameter size, the number of tokens per image, and the number of response turns, making fair comparison difficult. However, in response to your request, we will update this result in the paper.

---

### Author Response · Authors · 2025-11-25
**PDF and author response updated**

Dear reviewers,

We have updated the manuscript PDF to address the weaknesses and questions. Since the discussion period is approaching its end, we kindly ask you to review our responses and the revised version at your earliest convenience.

---

### Author Response · Authors · 2025-12-04
**Summary of the discussion**

For the convenience of all readers, here we briefly summarize the reviewers’ concerns and how we addressed them:

1. Reward design & ablations.

**Reviewers 3Vfh and Rz9J** asked for further details and ablation studies of our reward design. In response, **we added a section ablating each component of our reward**, and introduced that our reward was carefully tested and not set arbitrarily.

2. Clarification on the miswording of “Reward Hacking.”

**Reviewer rgJo** is concerned about the issue of “Reward Hacking”, which manifests as duplicate content being generated in EGO subtasks. **We clarified that this was a miswording**, the real issue should be a generalization problem that only occurs on the [EGO] subtask, and can be solved by deliberate data collection.

3. Generating "NO REPLY" each time when the model does not need to reply hinders inference efficiency.

**Reviewer 3Vfh** raised that "using 'NO REPLY' text token ... a complete generation process (generating both tokens) is still required, which limits its inference efficiency", and **Reviewer Rz9J** wondered the actual inference speed.
**We added an experiment reporting actual wall-clock latency, which is comparable to other baselines. We also discussed a token-efficient reply time decision scheme in our paper, where we explained that the current approach facilitates adaptation to mainstream frameworks.**

4. Broader Evaluation & Baselines.

**Reviewer 3Vfh** suggested that we should compare with more baselines. **We explained that we have compared with all baselines we can implement.** Other models do not have available open-source code, nor do they report proactive metrics.

We acknowledge that this work is not without limitations, and we have made every effort to respond to all reviewer questions and concerns. **We sincerely hope that the community can consider both the novelty of our contributions and the importance of the problem we aim to address**, i.e. enabling multimodal large models to operate effectively in real-time, live-streaming scenarios. We would be grateful if our paper could be given the opportunity to reach a wider audience.

Sincerely,

The Authors

---

### Meta-Review · Area_Chair_LNYN · 2026-01-06

**Summary:**

This paper investigates the integration of GRPO, into Video MLLMs to enable proactive, real-time interaction in streaming scenarios. All reviewers agree that this research direction is timely and of significant importance moving toward more natural, live-streaming AI agents.

The authors successfully addressed initial critical concerns regarding reward design, inference efficiency, and "reward hacking." Specifically, the new ablation studies proved the necessity of the four-component reward mechanism, and the wall-clock latency tests confirmed the model's practical utility. While a "generalization gap" was noted for long-form ego-centric videos, the authors clarified that this is a data-collection challenge rather than a flaw in the methodology. Given the strong rebuttal and the resolution of major technical objections, the AC believe that the reviewers would have reached a positive consensus if the discussion continued and thus recommend acceptance of the paper.

**Reviewer Concerns:**

Most critical reviews are resolved.
- Reward Design and Ablations (Reviewers 3Vfh, Rz9J): The reviewers requested evidence that the four reward components were necessary. The authors added a new ablation study showing that without repetition and density, the model falls into a loop of generating duplicate responses to "inflate" its score or responds at every frame, making inference unfeasible.
- Misinterpretation of "Reward Hacking" (Reviewer rgJo): The reviewer was concerned that the model was exploiting the reward function by repeating content. The authors clarified this was a generalization issue specific to long-form ego-centric videos rather than a fundamental flaw in the RL alignment. They corrected the wording from "reward hacking" to "generalization gap" in the revised manuscript.
- Inference Efficiency of "NO REPLY" (Reviewer 3Vfh): Reviewers argued that generating a text token to stay silent is inefficient. The authors demonstrated through wall-clock latency tests that the latency is comparable to baselines.

**Reviewer Scores:**

- Reviewer Rz9J: This reviewer is the most positive, praising the novelty of using GRPO for online video settings. Given the reward ablations and the inference speed report provided in the rebuttal, it is likely for the reviewer to keep the positive score.
- Reviewer rgJo: This reviewer was highly concerned about "reward hacking." The authors' clarification that this was a data-driven generalization issue rather than an RL failure should theoretically shift this score toward the positive side.
- Reviewer 3Vfh: This reviewer had the most technical questions (innovation of text-to-text, baseline comparisons, reward ablations). The authors provided a very strong response, including the additional ablation study that proves the necessity of their reward components. This reviewer's score would likely increase if they accept the constraints on open-source baseline comparisons.

---

### Decision · Program_Chairs · 2026-01-26

Accept (Poster)